# Centrally expressed Cav3.2 T-type calcium channel is critical for the initiation and maintenance of neuropathic pain

Sophie L Fayad[1], Guillaume Ourties[2,3], Benjamin Le Gac[1], Baptiste Jouffre[2,3], Sylvain Lamoine[2,3], Antoine Fruquière[4], Sophie Laffray[4], Laila Gasmi[1], Bruno Cauli[1], Christophe Mallet[2,3], Emmanuel Bourinet[4], Thomas Bessaih[1], Régis C Lambert[1]*, Nathalie Leresche[1]

[1]Sorbonne Université, CNRS, INSERM, Neurosciences Paris Seine – Institut de Biologie Paris Seine, Paris, France; [2]Université Clermont Auvergne, Inserm, U1107 Neuro-Dol, Pharmacologie Fondamentale et Clinique de la Douleur, Clermont-Ferrand, France; [3]ANALGESIA Institute, Faculty of Medicine, Clermont-Ferrand, France; [4]Institut de Génomique Fonctionnelle, Université de Montpellier, CNRS, INSERM, Montpellier, France

**Abstract** Cav3.2 T-type calcium channel is a major molecular actor of neuropathic pain in peripheral sensory neurons, but its involvement at the supraspinal level is almost unknown. In the anterior pretectum (APT), a hub of connectivity of the somatosensory system involved in pain perception, we show that Cav3.2 channels are expressed in a subpopulation of GABAergic neurons coexpressing parvalbumin (PV). In these PV-expressing neurons, Cav3.2 channels contribute to a high-frequency-bursting activity, which is increased in the spared nerve injury model of neuropathy. Specific deletion of Cav3.2 channels in APT neurons reduced both the initiation and maintenance of mechanical and cold allodynia. These data are a direct demonstration that centrally expressed Cav3.2 channels also play a fundamental role in pain pathophysiology.

*For correspondence:
regis.lambert@sorbonne-universite.fr

Competing interest: The authors declare that no competing interests exist.

## Editor's evaluation

The manuscript shows an important role for Cav2.3 channels in SNI-mediated allodynia and the firing properties of PV-expressing APT neurons. Mechanisms that underlie adaptations in chronic pain models are extremely important for the development of novel therapeutics for chronic pain and this could be a significant contribution in that regard.

## Introduction

Neuropathic pain is a wide spread condition with severe side effects and limited effectiveness of available treatments. Such public health issue stimulates the search for putative molecular targets, among which ion channels expressed in pain pathways are prime candidates (*Bennett and Woods, 2014*; *Bourinet et al., 2016*). In particular, at the peripheral level, it was shown that the Cav3.2 isoform of the low-threshold T-type calcium channels (T channels) is overexpressed in dorsal root ganglion (DRG) neurons after chronic constrictive injury of the sciatic nerve (*Jagodic et al., 2008*). Silencing the Cav3.2 isoform using intrathecal antisense oligonucleotides administration had profound antihyperalgesic and antiallodynic effects in various neuropathic pain models (*Bourinet et al., 2005*; *Messinger et al., 2009*; *Takahashi et al., 2010*). Furthermore, specific knockout (KO) of Cav3.2 in a subtype of DRG neurons alleviated both mechanical and cold allodynia produced by spared nerve

injury (SNI) (*François et al., 2015*). While these data converge toward a critical role for peripherally expressed Cav3.2 in neuropathic pain, very little is known on a putative pronociceptive role of centrally expressed Cav3.2.

In the central nervous system, in situ hybridization studies revealed strong expression of the Cav3.2 transcript in scattered neurons of the anterior pretectum (APT) (*Talley et al., 1999*), a crossroads of ascending and descending connectivity of the somatosensory system (*Rees and Roberts, 1993*). The idea that APT plays a key role in nociception dates back to the 1980s, when it was shown that brief low-intensity electrical or chemical stimulations of this nucleus elicited antinociception (*Prado and Roberts, 1985*; *Rees and Roberts, 1993*; *Roberts and Rees, 1986*). Involvement of APT in chronic pain was later confirmed by lesions and stimulations performed in different rodent pain models (*Rees et al., 1995*; *Rossaneis et al., 2014*; *Rossaneis and Prado, 2015*; *Villarreal et al., 2004*; *Villarreal et al., 2003*). To date the excitability of this structure has been little studied (*Bokor et al., 2005*) but an increase firing of fast-bursting neurons was reported in a model of central pain syndrome (*Murray et al., 2010*). Since T channels are involved in burst generation in numerous neuronal populations (*Lambert et al., 2014*), we took advantage of a murine model that allows the identification of the Cav3.2 channels and their conditional deletion in specific area (*François et al., 2015*) to investigate their role in shaping APT neuron excitability and the initiation and maintenance of neuropathic pain.

We show that (1) Cav3.2 channels are specifically expressed in GABAergic parvalbumin (PV)-expressing neurons of the APT; (2) bursting activity of these neurons is increased in the SNI model of neuropathic pain; (3) the Cav3.2 channel contribution to the bursting activity of the PV-expressing neurons is enhanced in the SNI model, and (4) their specific deletion in the APT significantly reduces both the initiation and maintenance of neuropathic mechanical and cold allodynia in the SNI model.

## Results

### Cav3.2 channels are expressed in PV-positive neurons of the APT

To examine Cav3.2 expression in APT neurons, we used the Cav3.2-eGFPflox (KI) mouse line, in which an ecliptic GFP tag is expressed in the extracellular loop of the Cav3.2 channel (*François et al., 2015*). Anti-GFP labeling revealed the expression of the Cav3.2-GFP fusion protein in a scattered subpopulation of neurons from naïve KI animals (*Figure 1A, B*). Co-labeling of GFP- and NeuN-positive cells (*Figure 1—figure supplement 1*) showed that 20.0 ± 3.9% of the APT cells express the Cav3.2-GFP channel. Since a previous study performed by *Bokor et al., 2005* in rats suggests that APT fast-bursting neurons strongly expressed PV, the overlap between Cav3.2-GFP+ and PV-positive (PV+) populations was estimated using GFP and PV co-labelings (*Figure 1—figure supplement 1*). We thus determined that 87.1 ± 10.0% of Cav3.2-GFP+ cells coexpress PV. Conversely 91.8 ± 5.6% of PV+ cells coexpress Cav3.2-GFP confirming the overlap between the Cav3.2-GFP+ and PV+ cell populations (*Figure 1C*).

### Burst firing of PV+ neurons is increased in SNI animals

Based on this overlap, we next investigated the firing activity of this subpopulation in anesthetized animals, using PV-Cre:Ai32 mice that selectively express channelrhodopsin-2 (ChR-2) in PV+ neurons. One to two tetrodes were lowered to the APT in the contralateral side of the sham or SNI surgeries to record multiunit spiking activity alongside with EEG signal (*Figure 2A, B*). Spike-sorting algorithms were further used to isolate one to three single-unit spiking activities per tetrode (*Figure 2C*). In sham mice using Photo-assisted Identification of Neuronal Population (PINP; *Lima et al., 2009*), PV+ single units were identified by the reliable short latency (5.3 ± 2.4 ms) evoked response consisting of one or more spikes elicited by each 470-nm blue-light pulse (*Figure 2D*). As shown in *Figure 2E*, 77% (*n* = 21) of the units had a peak around 2–5 ms in their autocorrelogram, indicating their ability to elicit bursts, and are thus referred to as 'bursting cells'. Their mean firing rate was 8.4 ± 2.8 Hz and bursts, consisting of 2–3 action potentials (mean: 2.9 ± 0.6) occurring at 252 ± 18.7 Hz, represented 13.7 ± 6.8% of the total number of spikes. The six remaining units showed a flat distribution of interspike intervals (ISIs) with a mean firing rate of 4.4 ± 1.7 Hz, and are thus referred to as 'regular cells' (*Figure 2—figure supplement 1*).

The spiking properties of PV+ neurons were next compared between sham and SNI animals. Although the proportion of bursting cells was similar in the two conditions (77% vs. 75%, respectively),

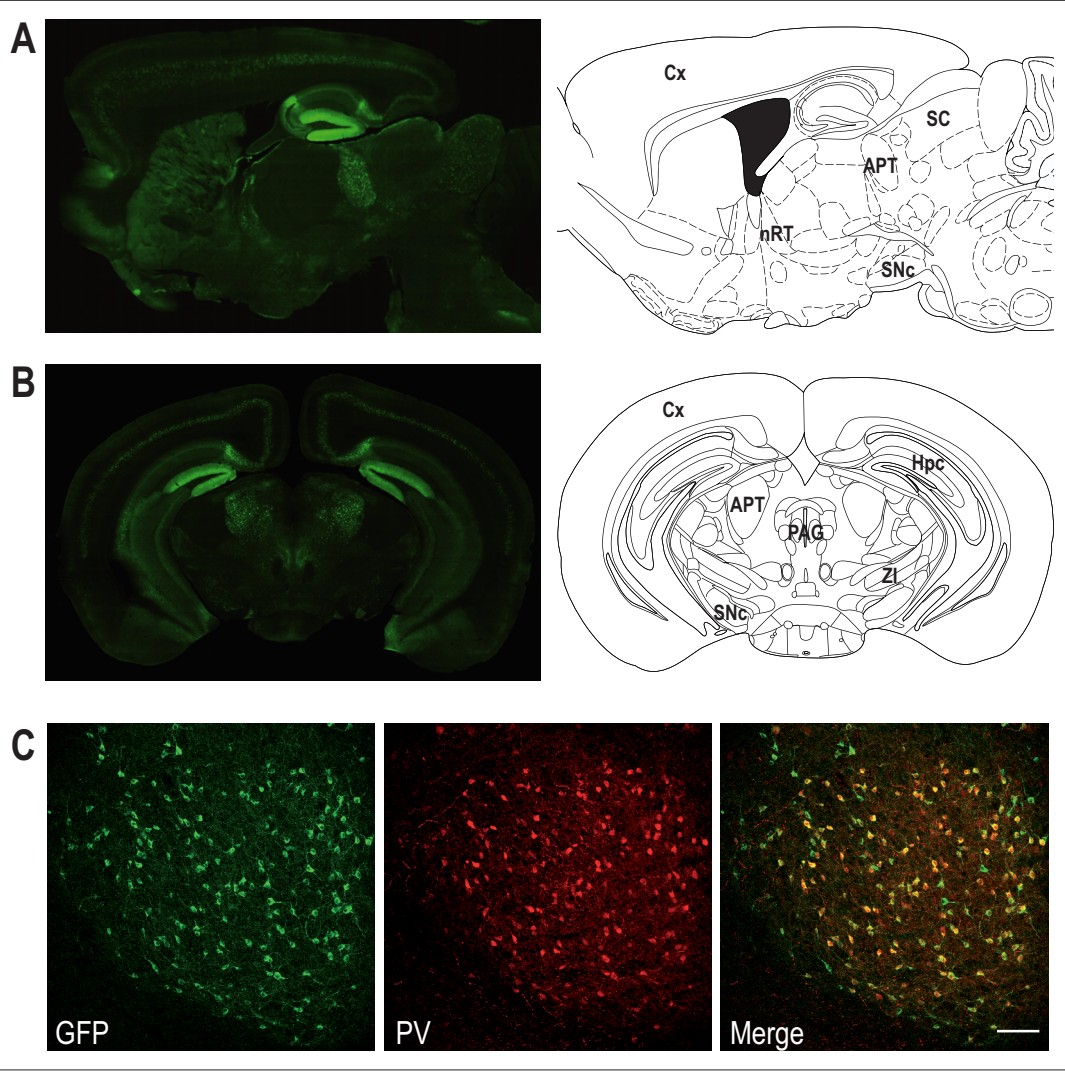

**Figure 1.** Coexpression of Cav3.2 and parvalbumin (PV) in APT neurons. Left panels: Cav3.2-GFP immunostaining on a parasagittal (**A**) and a coronal (**B**) section of KI mice brains. Right panels: corresponding Mouse Brain Atlas slides from Paxinos and Franklin (parasagittal: 1.08 mm lateral from Bregma; coronal: 2.8 mm posterior from Bregma). APT: anterior pretectum; Cx: cortex; Hpc: hippocampus; nRT: nucleus reticularis thalami; PAG: periaqueductal grey; SC: superior colliculi; SNc: substantia nigra pars compacta; ZI: zona incerta. (**C**) Confocal microscopy images of a coronal KI mouse brain section of the APT with GFP (green) and PV (red) co-labeling. Scale bar: 100 μm.

The online version of this article includes the following source data and figure supplement(s) for figure 1:

**Source data 1.** Quantification of Cav3.2-GFP- and parvalbumin (PV)-expressing neurons.

**Figure supplement 1.** Anterior pretectum (APT) neurons expressing GFP and parvalbumin (PV).

we observed a significant increase upon SNI, in the mean firing rate (*Figure 2F*; left panel; 11.4 ± 3.2 Hz; Wilcoxon sum rank test; p = 0.0036), in the proportion of spikes belonging to a burst (*Figure 2F*; middle panel; 20.9 ± 7.8%; Wilcoxon sum rank test; p = 0039), and in the mean spike frequency within a burst (*Figure 2F*; right panel; 270.6 ± 20.6 Hz; Wilcoxon sum rank test; p = 0.01). No difference was observed in the mean firing rates of regular cells for which the mean spike frequency in SNI animals was 6.7 ± 3.9 Hz (Wilcoxon sum rank test; p = 0.35).

We therefore concluded that most of the PV+ and Cav3.2+ neurons of the APT are bursting neurons and that their bursting activities are enhanced in the neuropathic pain state.

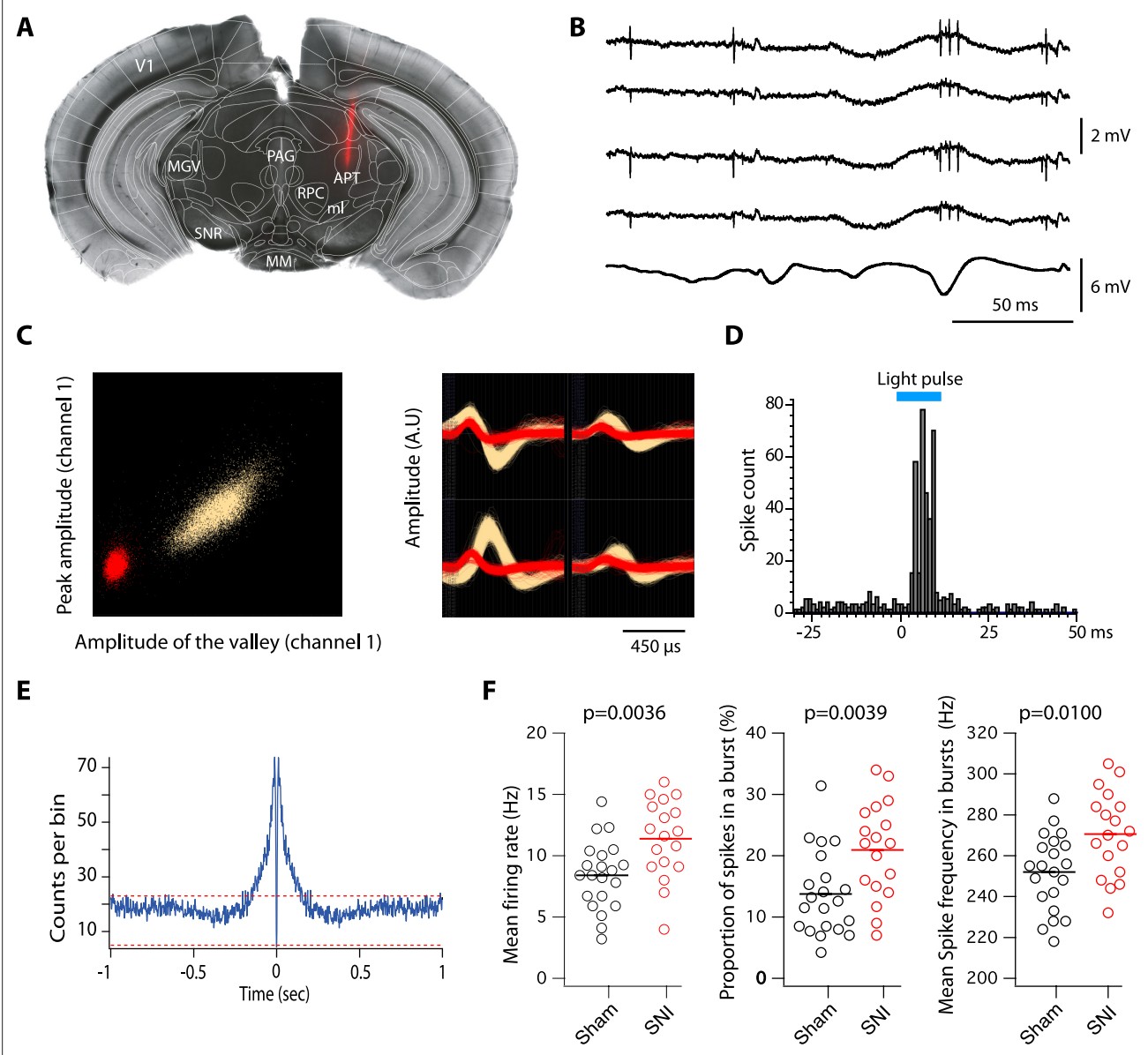

**Figure 2.** Impact of spared nerve injury (SNI) in PV+ anterior pretectum (APT) neurons. (**A**) Example image of a DiI track left by a recording electrode inserted into the APT. Red: DiI. (**B**) Raw signals from the four wires of a tetrode. Bottom trace shows the simultaneous EEG recording. (**C**) Left panel: example of a tetrode recording where two units were isolated. All detected action potentials are plotted as their waveform's amplitude from channel 1 versus the amplitude of their waveform's valley from channel 1 (in arbitrary units, A.U.). Using this feature space arrangement, two single-unit clusters were isolated (in red and yellow). Right panel: superimposed color-coded action potential waveforms captured by each recording site of the tetrode are shown for the two identified single units (the polarity of the signals is inverted). (**D**) Example of peristimulus time histograms illustrating spiking response to optogenetic stimulation (10 ms long, represented in blue) over 100 trials of a unit categorized into the PV+ category. (**E**) Autocorrelogram of recorded single unit for one example cell. 1 ms bins were used. Red dotted lines represent 99% confidence intervals. (**F**) Scatter dot plots of mean firing rate (left panel), proportion of spikes within a burst (middle panel), and mean spike frequency within a burst (right panel) for fast-bursting APT cells recorded in sham (n = 21 cells, 5 animals) and SNI (n = 18 cells, 4 animals) mice. p values for statistical comparisons were obtained using Wilcoxon sum rank test.

The online version of this article includes the following source data and figure supplement(s) for figure 2:

**Source data 1.** Firing properties of parvalbumin (PV)-expressing neurons after spared nerve injury (SNI).

**Figure supplement 1.** Anterior pretectum (APT) regular spiking neurons.

**Figure supplement 1—source data 1.** Firing properties of PV+ regular spiking neurons.

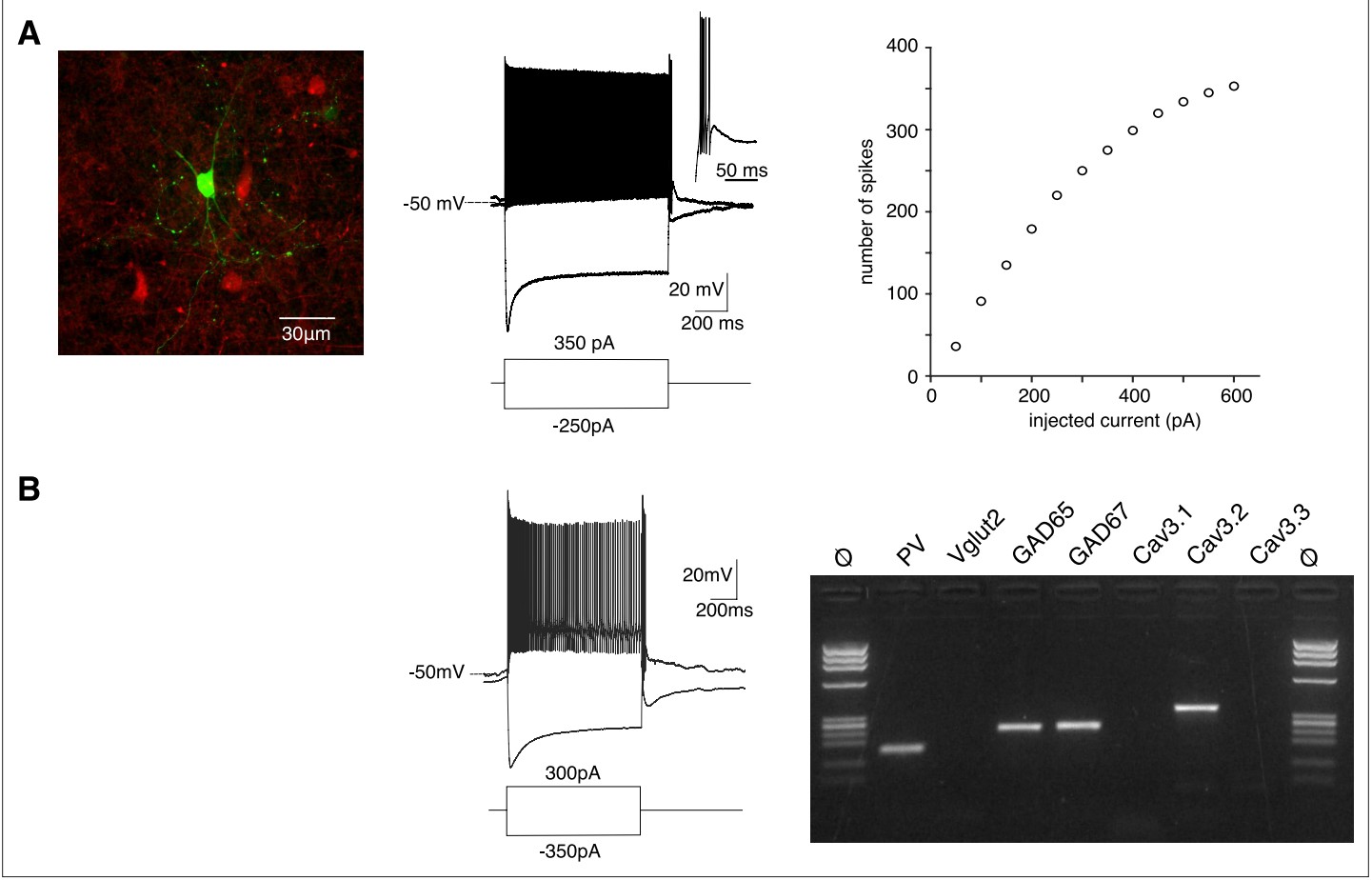

**Figure 3.** PV+ neurons expressing Cav3.2 channels are GABAergic fast-spiking neurons. (**A**) Left image: recorded neuron filled with biocytin (green). Neurons in red are nonrecorded PV+ neurons. Middle traces: depolarizing current injection evokes characteristic fast-spiking activity. Hyperpolarizing current injection evoked a pronounced sag and a rebound high-firing burst upon repolarization. Inset: enlargement of the bursting activity. Right graph: number of spikes evoked by 1 s long depolarizing current injection of increasing amplitudes. (**B**) Typical example of activities and scRT-PCR products observed in a PV+ neuron. Note the expression of the Cav3.2 channel in the GAD65- and 67-positive neuron.

The online version of this article includes the following source data for figure 3:

**Source data 1.** Excitability properties of PV+ neurons.

**Source data 2.** Original file of the full raw unedited gel shown in *Figure 3B*.

**Source data 3.** Molecular profile of PV+ neurons.

## Contribution of Cav3.2 channels to PV+ APT neuron excitability

In order to estimate how Cav3.2 channels contribute to the firing activity observed in vivo, we characterized the excitability of the PV+ neurons using whole-cell patch-clamp recordings combined with biocytin labeling in slices obtained from PV-Cre:AI14 mice that express TdTomato in PV+ neurons. In response to a pulse of depolarizing current all APT fluorescent neurons behaved as fast-spiking neurons with a high discharge rate (mean maximal firing rate upon depolarizing current of increasing amplitude: 214 ± 112 Hz, *n* = 60, *Figure 3A*). In response to the injection of a hyperpolarizing current step (peak hyperpolarization ranging from −95 to −105 mV), common features were a large amplitude sag (17 ± 6 mV, *n* = 62), indicating the presence of an Ih current, and a rebound depolarization that underlie a burst of high-frequency action potentials in 57 out of 62 recorded neurons (mean number of spikes 7 ± 4, *n* = 57).

Transcriptomic characterization of PV+ neurons was further performed by combining patch-clamp recordings and multiplex single-cell reverse transcriptase PCR (scRT-PCR; *Figure 3B*). All neurons expressed PV transcript and among them, 73.9% expressed GAD65 and/or GAD67 mRNA (*n* = 17/23), the remaining ones (*n* = 6/23) expressed Vglut2 mRNA. Importantly the Cav3.2 transcript was

present in 13 out of the 17 GABAergic neurons but never detected in glutamatergic neurons. mRNA of the two other Cav3 isoforms was also detected, albeit less frequently (Cav3.1: $n$ = 10/13; Cav3.3: $n$ = 3/13) in GABAergic neurons coexpressing Cav3.2 and in glutamatergic neurons (Cav3.1: $n$ = 5/6; Cav3.3: $n$ = 1/6). These results strongly suggest that PV+ neurons expressing the Cav3.2 channels are fast-spiking GABAergic neurons.

We then investigated whether the Cav3.2 isoform contributed to the rebound depolarization and its associated high-frequency burst firing. As shown in *Figure 4A, C*, application of 100 µM $Ni^{2+}$, a concentration that specifically blocks Cav3.2 channels (*Lee et al., 1999*), significantly decreased the number of action potentials (ctr: 7.0 ± 3.1/$Ni^{2+}$: 3.8 ± 2.8; $n$ = 8; Wilcoxon signed rank test p = 0.0039), the firing frequency (min ISI: ctr: 3.4 ± 0.9 ms/$Ni^{2+}$: 6.6 ± 4.0 ms; $n$ = 6; Wilcoxon signed rank test p = 0.0156) and the amplitude of the underlying depolarization measured in the presence of TTX (ctr: 13.6 ± 5.2/$Ni^{2+}$: 10.8 ± 6.1 mV; $n$ = 10; Wilcoxon signed rank test p = 0.0029). A block of all T channels isoforms using the pan antagonist TTA-P2 produced an even stronger decrease of the rebound activity, as expected from the results of the scRT-PCR (*Figure 4B, C*).

Since in vivo recordings showed an increased in bursting activities of PV+ neurons in SNI mice compared to sham control mice, we further compared the rebound bursts in slices from sham-operated and SNI mice. As shown in *Figure 4D*, the distribution of the maximal number of spikes of rebound bursts evoked in PV+ neurons is shifted toward larger values in SNI compared to sham-operated mice (mean maximal number of spikes: 9.1 ± 2.3, $n$ = 15, 6.3 ± 1.8, $n$ = 12, for SNI and sham, respectively; Wilcoxon sum rank test p = 0.0012). Importantly, this difference in distribution is suppressed when Cav3.2 channels are blocked with 100 µM $Ni^{2+}$, strongly suggesting that these channels participate in the increased rebound burst activities observed in SNI mice.

## Impact of Cav3.2 channel deletion in APT on mechanical and cold sensitivity

The Cav3.2-eGFPflox mouse line not only allows the visualization of Cav3.2 channels but also their deletion (*Figure 5—figure supplement 1*). Considering the in vitro results suggesting that Cav3.2 channels may contribute to the increased burst activities observed in PV+ neurons of SNI mice recorded in vivo, we investigated whether local deletion of these channels in APT could alleviate the neuropathic phenotype. Mechanical and cold allodynia, the cardinal somatosensory phenotypes typical of the SNI model were thus compared between mice that were bilaterally injected in the APT with either an AAV-Cre-mCherry virus (APT-KO mice; $n$ = 6 males and 6 females) or an AAV-mCherry (control KI mice; $n$ = 7 males and 9 females; *Figure 5A*) and then 2 weeks later subjected to the SNI surgery to induce the neuropathy. Mechanical sensitivity was assessed by measuring paw withdrawal threshold (PWT) on the operated hindpaw using Von Frey filaments (*Chaplan et al., 1994*). Prior to SNI procedure, two baseline measurements were performed, before and 2 weeks after viral injections. No significant differences were found in the PWT between these two measurements within each group or between KI and APT-KO mice, indicating that neither the viral injection by itself, nor the local deletion of Cav3.2 impacts acute mechanical sensitivity. Baseline measurements were therefore averaged within each group (*Figure 5B*). A statistically significant decrease in PWT was observed in KI mice following SNI from day 14 postsurgery, confirming the development of a mechanical allodynia. Importantly, this decrease was strongly attenuated in APT-KO compared to KI mice (*Figure 5B* and *Figure 5—figure supplement 1B*). This difference between APT-KO mice and their KI littermates was not due to motor or coordination deficits (*Figure 5—figure supplement 1C, D*).

Cold sensitivity was also assessed using the paw immersion test (*Figure 5D*). While KI mice showed a decrease in withdraw latency after SNI when the operated paw was immersed in a water bath at 18°, indicating the development of a cold allodynia, this phenotype was abolished in the preventive APT-KO mice. Altogether, these results clearly indicate that APT-Cav3.2 channels contribute to the initiation of mechanical and cold allodynia characterizing the SNI model in both male and female mice.

We next investigated whether these allodynic SNI features could be reversed by a therapeutic KO strategy of APT-Cav3.2. We thus injected the Cre-expressing virus 2 weeks after surgery, when allodynic symptoms reach their peak. Both mechanical and cold allodynia were rapidly rescued since tactile threshold lowering was drastically reduced (*Figure 5C*) and paw withdrawal latencies returned to preoperative baseline values (*Figure 5E*).

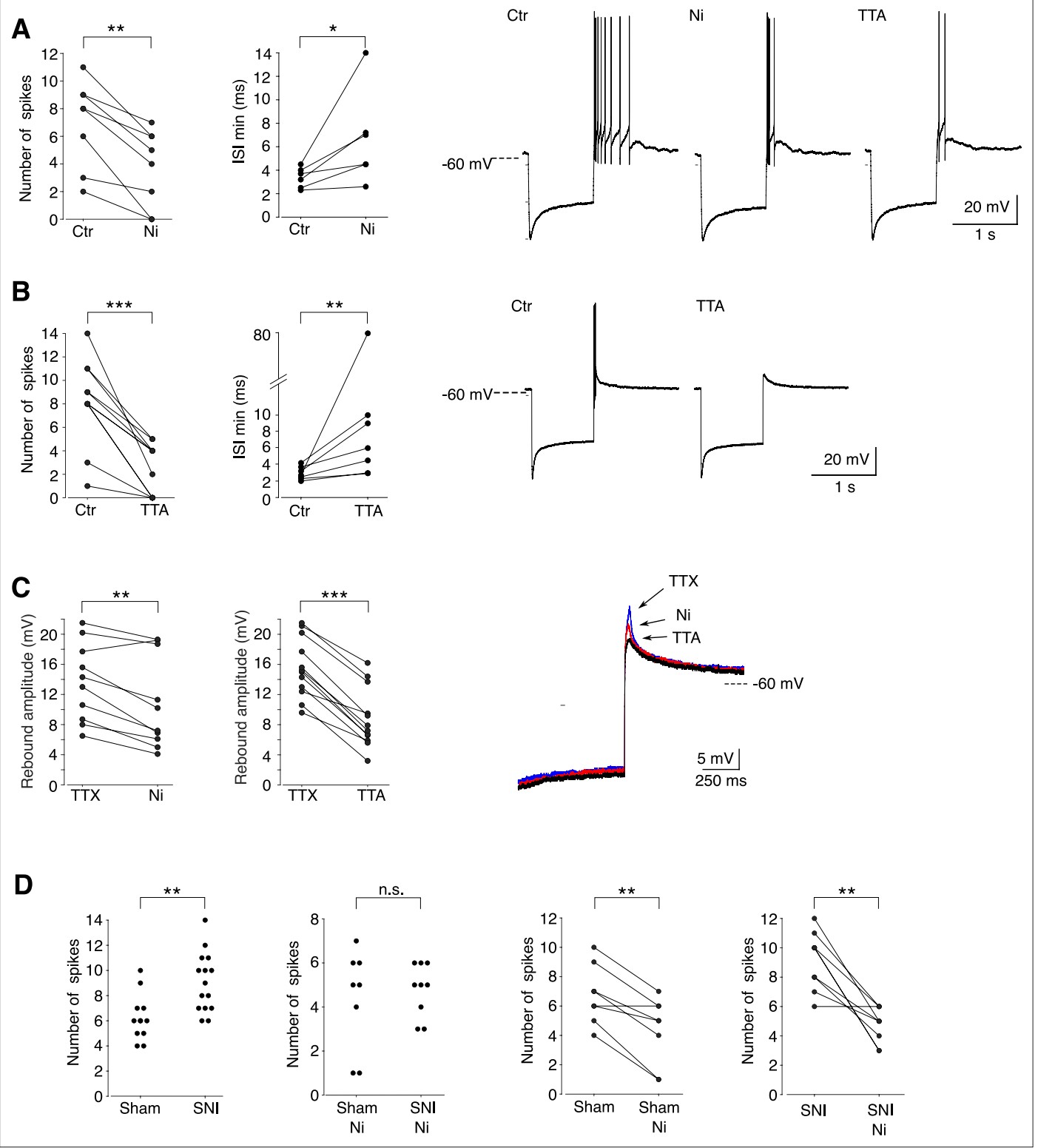

**Figure 4.** Cav3.2 channel contribution to the bursting activity of PV+ neurons is enhanced in spared nerve injury (SNI) mice. (**A**) Effect of 100 µM Ni[2+] applications on the number of spikes (left graph, $n = 8$) and the minimal interspike intervals (ISIs; right graph, $n = 6$) of the rebound bursts. Typical examples of these pharmacological effects are shown on the right. (**B**) Effect of 1 µM TTAP2 applications on the number of spikes (left graph, $n = 11$) and the minimal ISIs (right graph, $n = 7$) of the rebound bursts. Typical examples of these pharmacological effects are shown on the right. (**C**) Effect of 100 µM Ni[2+] (left graph, $n = 10$) and 1 µM TTAP2 (right graph, $n = 12$) applications on the amplitude of the rebound depolarization observed in the presence of 0.5 µM TTX. A typical example is presented in superimposed traces presented on the right. (**D**) The two left graphs compare the maximal

*Figure 4 continued on next page*

*Figure 4 continued*

number of spikes of the rebound bursts evoked in neurons of sham-operated and SNI mice in control condition (n = 12 and 15, respectively) and after application of 100 µM Ni²⁺ (n = 8 and 9, respectively). The effects of Ni²⁺ application on each neuron are presented in the two right graphs (n = 8 and 9 for sham and SNI, respectively). A, B, C, D right graphs: Wilcoxon signed rank test; D left graphs: Wilcoxon sum rank test. ***p < 0.001 ; **p: 0.001 < p < 0.01 ; *0.01 < p < 0.05.

The online version of this article includes the following source data for figure 4:

**Source data 1.** Ni and TTA effects on burst properties of PV+ neurons.

**Source data 2.** Ni and TTA effects on the rebound depolarization in PV+ neurons.

**Source data 3.** Burst properties of PV+ neurons in slices from sham and spared nerve injury (SNI) mice.

Altogether, these results show that Cav3.2 expressed in the APT is not only involved in the initiation but also in the maintenance of neuropathic phenotype in the SNI model.

## Discussion

We showed that deletion of Cav3.2 channels expressed in a GABAergic and PV-expressing subpopulation of APT neurons leads to an antiallodynic effect in the SNI neuropathic pain model.

Our in vitro data indicate that 92% of APT-PV+ neurons are able to discharge bursts of action potentials at high frequency underpinned by a rebound large transient depolarization due to the activation of T channels. In contrast to the well-recognized role played by the Cav3.1 and 3.3 isoforms of the T channels (*Kim et al., 2001*; *Lee et al., 2014*; *Pellegrini et al., 2016*), little evidences exist for the involvement of the Cav3.2 isoform in burst generation in the central nervous system (*Candelas et al., 2019*; *Dumenieu et al., 2018*). Here, the specific blockade of this isoform greatly reduced the amplitude of the depolarization underlying burst generation and decreased the number and frequency of action potentials within a burst. Therefore, although our transcriptomic and pharmacological results suggest that APT-PV+ GABAergic neurons express multiple isoforms of the T channels, we demonstrate that the Cav3.2 isoform has an essential contribution to the bursting behavior in this subpopulation of APT neurons and its increase in SNI mice.

Available data about the excitability of APT neurons are very limited. *Bokor et al., 2005* reported a heterogeneous firing pattern in rat APT neurons with about 25% of them presenting high-frequency discharges. Although tested on only three neurons, these fast-bursting neurons were described as strongly expressing PV. Deciphering the roles of a structure that displays such heterogeneity clearly requires recordings of identified neuronal subpopulation. The use of the PINP method allowed us to perform such specific characterization and our in vivo recordings of APT-PV+ neurons showed that 77% of these neurons discharge in bursts with a clear enhancement of bursting activity in SNI mice. Investigating changes in neuronal spiking activity in spinal-lesioned rats, *Murray et al., 2010* also described an increased firing rate of APT neurons with a higher percentage of neurons exhibiting at least one burst compared with sham-operated controls. The increase in bursting behavior reported here was more pronounced than in *Murray et al., 2010* as not only the number of bursts but also the number of spikes and their frequency were enhanced indicating a profound change in this class of APT neurons in neuropathic condition. This disparity likely stems from our ability to record an identified subpopulation of APT neurons, although a difference between neuropathic models cannot be excluded. When considering other brain structures, modifications of high-frequency-bursting activities in pain condition have been reported in human, as well as in various experimental models, and this is particularly well documented in the thalamocortical system. In awake patients with neurogenic pain, recordings from thalamic neurons show an increase in the occurrence of high-frequency bursts (*Jeanmonod et al., 1996*; *Lenz et al., 1989*; *Llinás et al., 1999*). Similar higher proportions of bursting neurons have been reported in ventralis postero-lateralis thalamic neurons following contusive spinal cord lesions in rats (*Gerke et al., 2003*) and monkeys (*Weng et al., 2000*). Furthermore, in the same thalamic nucleus, an increase in the frequency of T-channel-dependent bursts was reported following induction of visceral pain in mice (*Kim et al., 2003*).

The increased burst firing in APT-PV+ neurons in neuropathic condition prompted us to investigate whether Cav3.2 channels may contribute to this phenotype. Our behavioral analysis firstly showed that removing Cav3.2 specifically from APT-PV+ neurons strongly reduced mechanical and cold allodynia

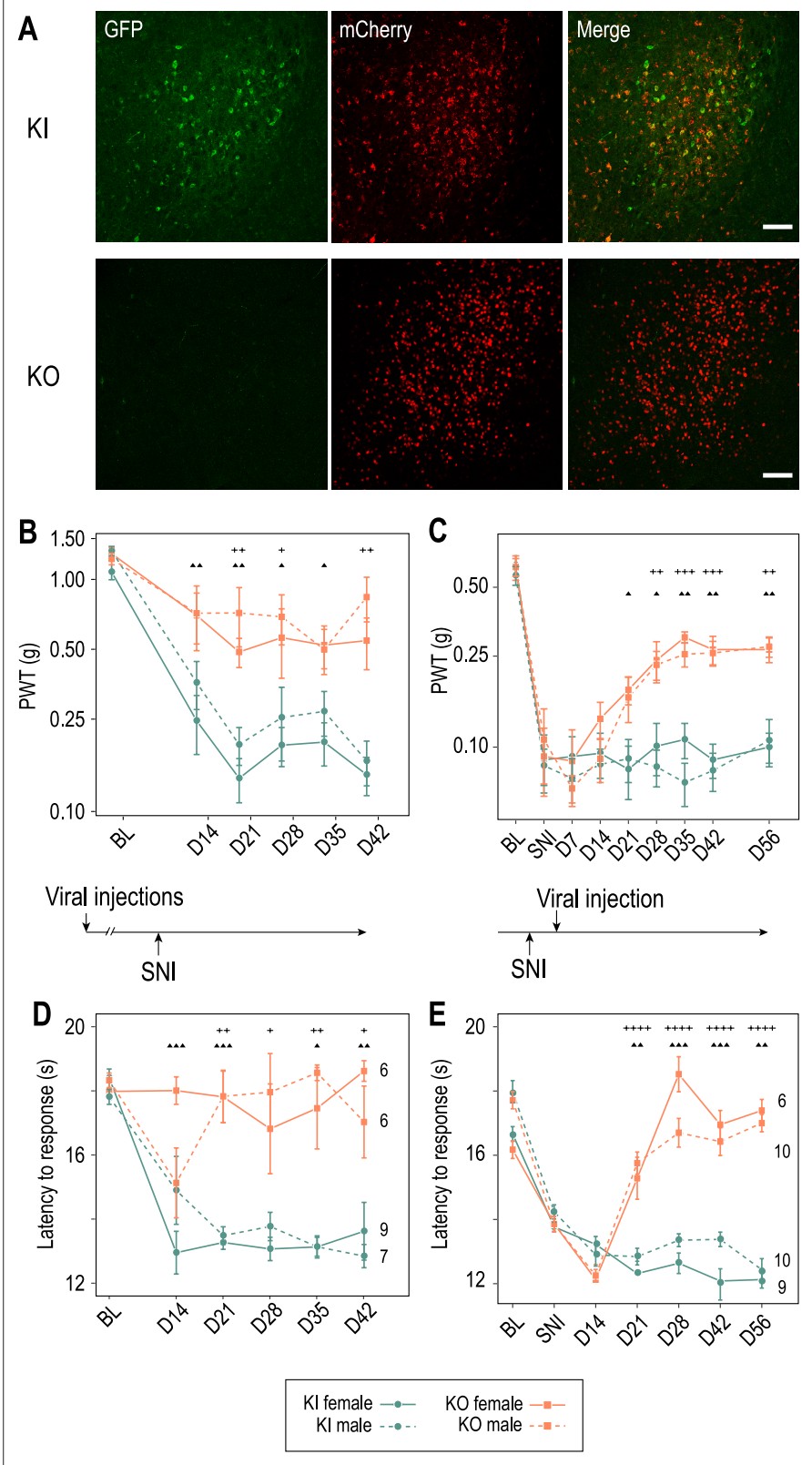

**Figure 5.** Cav3.2 preventive and therapeutic knockout in the anterior pretectum (APT) alleviates neuropathy induced by spared nerve injury (SNI). (**A**) Confocal microscopy images of GFP (green) and mCherry (red) co-labeling in the APT of KI-Cav3.2-GFP (Control) mice and KO-Cav3.2-APT (KO) bilaterally injected in the APT with AAV8-hSyn-mCherry and AAV8-hSyn-mCherry-CRE virus, respectively, and further tested for mechanical and cold

*Figure 5 continued on next page*

*Figure 5 continued*

sensitivity. Scale bar: 100 μm. Neuropathic behaviors were tested in male (dashed lines) and female (solid lines) mice with preventive (**B, D**) and therapeutic (**C, E**) KO of Cav3.2 in the APT (orange, ■), and in control KI mice (green, ●). (**B, C**) Mechanical sensitivity was assessed by measuring paw withdrawal thresholds (PWTs) in response to Von Frey filaments stimulations using the up-and-down method. (**D, E**) Cold sensitivity was assessed by measuring the paw withdrawal latency in response to immersion in 18°C water. For preventive KO (**B, D**) mice were tested prior to the SNI (BL) and once a week during the 6 subsequent weeks (days 14–48). For therapeutic KO (**C, E**) mechanical and cold sensitivity were assessed before (BL) and 14 days after SNI (SNI), and tested during several weeks following the subsequent viral injections (days 14–56 after viral injection). (**B–E**) Wilcoxon sum rank test for female KO versus KI and male KO versus KI comparison at each time point. ++++ : p < 0.0001, +++ or ▲▲▲: 0.0001 < p < 0.001, ++ or ▲▲: 0.001 < p < 0.01, + or ▲: 0.01 < p < 0.05 (males: +; females: ▲).

The online version of this article includes the following source data and figure supplement(s) for figure 5:

**Source data 1.** Characterization of the effect of APT-Cav3.2 preventive or curative knockout on neuropathic pain induced by spared nerve injury (SNI).

**Figure supplement 1.** Preventive Cav3.2 knockout in the anterior pretectum (APT) has no impact on motor coordination and spontaneous locomotion.

**Figure supplement 1—source data 1.** Characterization of preventive APT-Cav3.2 knockout effect on mechanical sensitivity and locomotion.

---

evoked by SNI. Furthermore, removing the Cav3.2 channels when the SNI effects were fully developed rescues the different allodynic symptoms indicating that Cav3.2 channels not only contribute to the development but also to the maintenance of the pain phenotype and strongly suggests their direct electrogenic implication in sustaining allodynia. Since these effects were observed on both male and female mice, the Cav3.2-dependent modification in APT-PV+ neurons activities sustaining allodynia is not sex specific, unlike what was recently reported for layer five PV+ neurons in the prelimbic region of the medial prefrontal cortex using the same neuropathic model (*Jones and Sheets, 2020*). Such lack of sex specificity suggests that this mechanism is a fundamental component of chronic pain processes.

Comparing the pain phenotypes of wild-type and global Cav3.2$^{-/-}$ mice, *Choi et al., 2007* reported reduced sensitivity to acute pain in KO mice but no change in the overall development of neuropathic pain. However, at the peripheral level, multiple data point toward Cav3.2 as a main actor in acute and chronic pain (*Bourinet et al., 2016*; *Bourinet et al., 2005*; *François et al., 2015*; *Jagodic et al., 2008*; *Messinger et al., 2009*; *Takahashi et al., 2010*) suggesting a possible compensatory phenomenon in condition of complete channel deletion. At the supraspinal levels, very little is known. In the chronic constriction injury model, an increase in Cav3.2 mRNA was observed in the anterior cingulate cortex and local injection of NNC 55-0396, a poorly selective T-channel antagonist (*Huang et al., 2004*; *Schäfer et al., 2016*), partially relieved allodynia (*Shen et al., 2015*). Using various poorly specific T-channel antagonists, *Chen et al., 2010* showed a T-channel-dependent activation of ERK in the paraventricular thalamus following acid-induced chronic muscle pain that was abolished in Cav3.2$^{-/-}$ mice. Furthermore, intracerebroventricular injection of the specific antagonist of the three T-channel isotypes, TTA-A2, induced analgesia blunting the effect of paracetamol, the most widely used remedy to treat mild pain, an effect also abolished in Cav3.2$^{-/-}$ mice (*Kerckhove et al., 2014*). However, such experiments only suggested a central pronociceptive role of Cav3.2 channels. Our results, showing that local deletion of Cav3.2 channels in 20% of the APT neurons drastically reduced allodynia, provide direct evidence of the involvement of central Cav3.2 channels in neuropathic pain.

Functionally, as APT-PV+ neurons project to the high-order somatosensory thalamic nucleus, posterior nucleus (PO) (*Bokor et al., 2005*; *Giber et al., 2008*), this antiallodynic effect could be a consequence of the direct involvement of this neuronal population in processing nociceptive afferents from the periphery. However, the net effect of the increased bursting activity of APT-PV+ GABAergic neurons on the PO activity under neuropathic conditions is difficult to predict. Although they account for only a small percentage of GABAergic projections impinging on zona incerta (ZI) (*Giber et al., 2008*), APT PV+ neurons project to the ZI, which in turn inhibit thalamic neurons (*Barthó et al., 2002*; *Lavallée et al., 2005*). Their increased bursting activity may therefore mediate disinhibition of PO neurons by silencing their ZI targets, while conversely mediating direct monosynaptic inhibition of PO neurons. Functionally, however, these two effects may not be opposed, as the impact of inhibitory input on thalamic neurons may result in a change in firing mode rather than in a clear decrease in

firing. Indeed, PO neurons, which highly express Cav3.1 T channels (*Talley et al., 1999*), are prone to generate rebound burst firing after transient hyperpolarization due to strong IPSP bursts originating from the APT neurons as shown in vitro by *Bokor et al., 2005*. However, *Masri et al., 2009* reported an increase in firing of the PO neurons but no change in their bursting activity in spinal-lesioned rats, but the recordings were restricted to a localized area of PO containing neurons that receive convergent inputs from both hind paw and vibrissae pad. Therefore, given the potential functional and anatomical regionalization of the PO (*El-Boustani et al., 2020*), determining the net effect of PV+ APT neuron activity on the thalamic neurons will required further in vivo recordings performed in their specific projection area.

Finally, anatomical and functional investigation of the APT described descending pathways impinging through complex routes on various spinal cord neuronal types and involved in pain processing (*Genaro et al., 2019*; *Rees and Roberts, 1993*; *Terenzi et al., 1992*; *Terenzi et al., 1991*; *Villarreal et al., 2004*). So far, it remains to be determined whether the PV+ neurons of the APT may participate to these descending pathways and modulate nociception at the spinal level.

In conclusion, our data point to Cav3.2 channels as a prime target for developing innovative analgesic pharmacology that will not only act at the peripheral level but also in central structures. Furthermore, they highlight the need to study the functional expression of channels in specific neuronal population to decipher the complex pain processing that occurs in the central nervous system.

## Materials and methods

### Lead contact and materials availability

For each figure, data are included in the figure – source data files.

Further information and requests for resources and reagents should be directed to and will be fulfilled by the Lead Contact, Régis C Lambert (regis.lambert@sorbonne-universite.fr).

### Experimental model and subject details

#### Animals

Cav3.2$^{eGFP-flox}$ (Cacna1h$^{tm1.1(epH)Ebou}$/J) knock in (KI) mouse line was obtained from Dr. E. Bourinet (*François et al., 2015*). Briefly, these mice express the ecliptic GFP (eGFP) in an extracellular loop of the Cav3.2 T channel. LoxP sites were inserted allowing the deletion of the Cav3.2-eGFP coding sequence by *cre*-recombinase (*cre*). Mouse lines of PV *cre* (B6.129P2-Pvalb$^{tm1(cre)Arbr}$/J, stock #017320) mice expressing *cre* under the PV promoter, Ai14 (B6.Cg-Gt(ROSA)26Sor$^{tm14(CAG-tdTomato)Hze}$/J, stock #007914) mice expressing tdTomato reporter, and Ai32 (B6.Cg-Gt(ROSA)26Sor$^{tm32(CAG-COPC4*H134E/EYPF)}$ $^{Hze}$/J, stock #024109) mice expressing channelrhodopsin-2 (ChR2)-eYFP were purchased from Jackson lab. PV-Cre:Ai14 and PV-Cre:Ai32 mice were obtained by crossing female PV-Cre with male Ai14 or Ai32 mice, respectively. Animals were housed in groups of 2–5 per cage, with a 12/12 hr light/dark cycle in a pathogen-free facility maintained at 22–24°C, and access to food and water ad libitum. All procedures complied with the ethical guidelines of the Federation for Laboratory Animal Science Associations (FELASA) and with the approval of the French National Consultative Ethics Committee for health and life sciences (authorization number: 17958).

### Method details

#### Immunocytochemistry and imaging

Mice (P28–P217) were anesthetized with 2% isoflurane, injected with a lethal dose of pentobarbital (150 mg/kg) and transcardiacally perfused with a 4°C solution of Artificial Cerebro-Spinal Fluid (ACSF), oxygenated with a mixture 95% $O_2$/5% $CO_2$, containing (in mM): 125 NaCl, 2.5 KCl, 2 $CaCl_2$, 1 $MgCl_2$, 1.25 $NaH_2PO_4$, 26 $NaHCO_3$, 25 glucose. The brains were then rapidly extracted from the skull and incubated in a solution of paraformaldehyde diluted to 4% in phosphate buffer solution (PBS: phosphate buffer saline) for postfixation overnight at 4°C. 40-µm-thick coronal slices were cut using a vibratome (Leica VT1000S) in PBS.

All immunohistochemical staining were performed at room temperature. The slices were rinsed three times in Tris buffer saline (TBS, containing 50 mM Tris Base and 150 mM NaCl) at pH 8.4, then incubated in TBS solution supplemented with 0.05% Tween 20, 0.2% Triton X-100 and 10% Donkey Serum (TBSTD) for 1h30. The slices were then incubated with the primary antibodies (see Table) in

TBSTD solution overnight, rinsed three times with TBS, incubated with secondary antibodies (see Table) in TBSTD for 2 hr and rinsed again three times before mounting between slide and coverslip in Fluoromount.

Whole slice images were acquired by epifluorescence microscopy under a macroscope (Axio Zoom, V16 Zeiss). Mosaics were made at ×200 magnification and processed with the ZEN software (Zeiss). The images used to study the coexpressions were acquired with confocal microscopy (Leica TCS SP5) at ×20 and ×63 objectives.

## SNI model

SNI or sham surgeries were performed as previously described (*Decosterd and Woolf, 2000*) under ketamine–xylasine anesthesia (100 and 10 mg/kg, respectively) on male and female KI mice (10–15 weeks old). The left thigh to be operated on was slightly elevated and an incision of about 1 cm is made between the hip and the knee. The muscles enclosing the sciatic nerve compartment were moved apart with a round-ended bent scissor. The common peroneal and tibial branches of the sciatic nerve were exposed and tightly ligated with 6-0 silk suture. A fragment of nerve was transected distally to the ligation. The sural branch was left intact. The muscles were then put back in apposition and the skin was sutured using 4-0 Vicryl (Ethicon). Mice were then kept in a 37°C warming chamber until recovery from anesthesia, before being returned to their homecages. In sham surgery, the sciatic nerve branches were exposed, but not injured.

## Surgery and preparation for in vivo electrophysiological recordings

Mice (males; 8–12 weeks; sham mice $n = 5$; SNI mice $n = 4$) were anesthetized with isoflurane vaporized in a mixture of oxygen and air (4% for induction, 1.5–2% during surgery). Body temperature was maintained at 37°C via a servo-controlled heating blanket and a rectal thermometer (Harvard Apparatus, Holliston, MA). Bupivacaine (subcutaneous) was administered in the regions to be incised 15 min prior to the first incision. Mice were placed in a stereotaxic apparatus and a craniotomy was made directly above the anterior pretectal nucleus (−2.7 to −3.0 mm A/P, 1–1.2 mm M/L). To minimize damage during electrode penetration, the dura was resected and the exposed surface was coated with a layer of silicon oil. After electrodes placement, isoflurane was decreased to 0.8–1%. If the animals presented any sign of discomfort, the percentage of isoflurane was increased. Electrodes were lowered to the anterior pretectal nucleus based on readings from the micromanipulator (depths: 2600–2900 µm).

## In vivo electrophysiological recordings

Animals underwent the procedure 2–3 weeks after SNI or sham surgeries. Recordings of units were obtained using quartz-insulated platinum/tungsten (90%/10%) tetrodes (~1–2 MOhm, Thomas Recording). Before insertion, the rear of the tetrodes was painted with fluorescent 1,1'-dioctadecyl-3,3,3',3'-tetramethyl indocarbocyanine perchlorate (DiI, 10% in ethanol, Invitrogen). As this dye is a lipophilic neuronal tracer, it allowed assessment of the recording depth. One to three tetrodes were guided independently at 1-µm resolution through a five-channel concentric microdrive head (Head05-cube-305-305-b, Thomas Recording GmbH) with 305 µm interelectrode spacing.

Raw signals were filtered (600–6000 Hz; Neuralynx recording systems), amplified (5000×), digitized at 33,657 Hz, and stored with stimulus markers (Cheetah 5 software; Neuralynx). Waveforms crossing set thresholds (300–500 µV) were captured via the A/D card and analyzed offline. At the end of the experiment, mice were injected with a lethal dose of euthasol, and the brains were removed and placed in a 4% paraformaldehyde solution for 48 hr. The brains were then transferred and stored in PBS (0.1 M). Sections (280 µm thick) containing the anterior pretectal nucleus were cut with a vibratome (Leica VT1000S). Sections were then mounted in Fluoromount medium.

## Slice preparation

Investigation of PV+ neuron firing properties were performed on slices obtained from 13 male and 11 female PV-Cre:AI14 mice (P23–P123, median: P35). Comparison of PV+ neuron-bursting properties between SNI and sham-operated mice were performed on slices prepared 22–41 days after surgery (Sham: six mice, four males, two females, P87–P103; SNI: seven mice, five males, two

females, P91–P110). Mice were anesthetized with isoflurane before decapitation. The brain was carefully removed and placed for a few minutes into a 4–8°C bicarbonate-buffered saline (BBS) solution containing (in mM): 125 NaCl, 2.5 KCl, 2 CaCl$_2$, 1 MgCl$_2$, 1.25 NaH$_2$PO$_4$, 26 NaHCO$_3$, and 25 glucose (osmolarity: 305 mOsm; pH 7.3 after equilibration with 95% O$_2$ and 5% CO$_2$). Coronal slices (250 μm) were then cut using a vibratome (Campden Instruments 700 SMZ2). The slicing procedure was performed in an ice-cold solution containing (in mM): 130 potassium gluconate, 15 KCl, 2 ethylene glycol-bis(ß-aminoethyl ether)-N,N,N',N'-tetraacetic acid (EGTA), 20 4-(2-hydroxyethyl)piperazine-1-ethanesulfonic acid (HEPES), 25 glucose, 1 CaCl$_2$, and 6 MgCl$_2$ (304 mOsm, pH 7.4 after equilibration). Slices were then transferred for few minutes to a solution containing (in mM): 225 D-mannitol, 2.5 KCl, 1.25 NaH$_2$PO$_4$, 25 NaHCO$_3$, 25 glucose, 1 CaCl$_2$, and 6 MgCl$_2$ (310 mOsm, 4–8°C, oxygenated with 95% O$_2$/5% CO$_2$), and finally stored for the rest of the experiment at 32°C in oxygenated BBS. For all recordings, slices were continuously perfused with oxygenated BBS at 32°C.

## In vitro whole-cell patch-clamp recording

Brain slices were screened for fluorescent neurons using a filter set that allowed us to detect dt-tomato fluorescence. Neurons were visualized and patched with borosilicate pipettes (resistance 3–5 MOhm). The intracellular solution contained (in mM): 140 potassium gluconate, 3 MgCl$_2$, 10 HEPES, 0.2 EGTA, and 4 disodium ATP (pH 7.3; 300 mOsm). For some experiments, biocytin (2 mg/ml) was added to the intracellular solution. Patch-clamp electrodes were connected to a AxoPatch 200B (Axon Instrument) amplifier. Protocols and acquisitions were controlled by the Clampex software (Molecular Devices). The membrane potentials were filtered by a 4-pole Bessel filter set at a corner frequency of 2 kHz and digitized online at a sampling rate of 20 kHz. The access resistance was 10–20 MOhm and was monitored throughout the experiment. Data were discarded if the access resistance changed by more than 15% during the experiment. Current clamp recordings were performed in the continuous presence of 10 μM CNQX and 1 μM SR95531 to suppress spontaneous synaptic activities. Only one neuron per slice was recorded in experiments requiring the application of T-channel antagonists.

At the end of the recordings, slices containing biocytin-filled neurons were fixed overnight by immersion in paraformaldehyde (4% in PBS 1 M) and then washed with 1 M PBS. After incubating the slices with Triton (0.2% in PBS 1 M) for 1 hr, biocytin-filled neurons were revealed using Streptavidin Alexa Fluor 488 (1:1000; 3 hr in dark; Invitrogen). Slices were then washed in PBS (1 hr) before being mounted on cover slides.

## Cytoplasm harvesting and scRT-PCR

For scRT-PCR, recordings were performed on slices obtained from four male and two female PV-Cre:AI14 mice (P15–P21). At the end of the whole-cell recording, lasting less than 15 min, the cytoplasmic content was aspirated in the recording pipette. The pipette's content was expelled into a test tube and reverse transcription was performed in a final volume of 10 μl, as described previously (*Lambolez et al., 1992*). The scRT-PCR protocol was designed to probe simultaneously the expression of Cav3 isotypes, GAD65/67, VGluT2, and PV. Two-step amplification was performed essentially as described (*Cauli et al., 1997*; *Devienne et al., 2018*). Briefly, cDNAs present in the 10 μl reverse transcription reaction were first amplified simultaneously using all external primer pairs listed in the Key Ressources Table. Taq polymerase and 20 pmol of each primer were added to the buffer supplied by the manufacturer (final volume, 100 μl), and 20 cycles (94°C, 30 s; 60°C, 30 s; 72°C, 35 s) of PCR were run. Second rounds of PCR were performed using 1 μl of the first PCR product as a template. In this second round, each cDNA was amplified individually using its specific nested primer pair (Key Ressources Table) by performing 35 PCR cycles (as described above). 10 μl of each individual PCR product were run on a 2% agarose gel stained with ethidium bromide using ΦX174 digested by *Hae*III as a molecular weight marker.

## Virus stereotaxic injections

Male and female KI (6–8 weeks) were anesthetized with a ketamine–xylazine mixture (100 and 10 mg/kg, respectively) and placed on a heating pad. A vitamin B12 eye drop (Twelve TVM) was applied to the eyes and a subcutaneous injection of sterile saline solution (NaCl 0.9%) was performed to prevent dry eyes and dehydration, respectively. The surgical area was cleaned with ethanol and sanitized with an iodine solution (Vetedine). Lidocaine was administered subcutaneously at the incision site. Mice

were placed on a stereotaxic apparatus and a craniotomy was performed over the area of interest. Saline solution was regularly applied to the skull.

Injection pipette was lowered to the APT coordinates (Bregma: −2.70 to −2.80 mm; Mediolateral: ±1.10 to 1.15 mm; Depth: −2.65 to −2.70 mm from the dura) and either, 1 µl of AAV8-hSyn-mCherry-Cre ($4.9 \times 10^{12}$ ppm) or 1 µl of AAV8-hSyn-mCherry ($4.6 \times 10^{12}$ ppm) were injected bilaterally with a graduated injection wheel (Narishige, 100 µl/rev) at a rate of 0.1–0.2 µl/min. Five to ten minutes after injection, the pipette was slowly raised. The wound edges were then put back in place and the skin sutured with Vicryl 4-0 thread (Ethicon) or surgical glue (Vetbond 3M). Mice were kept on a heating pad until recovery from anesthesia before being returned to their home cages.

## Behavioral tests

Two series of behavioral tests were conducted separately in two different laboratories by different experimenters in blinded condition. Results are presented independently in *Figure 5* and *Figure 5— figure supplement 1*. Mechanical and cold sensitivity was tested in preventively injected KI and KO mice, as well as in therapeutically injected KI and KO SNI mice in Clermont-Ferrand. Mechanical sensitivity in preventively injected KI and KO mice was also assessed in Paris, along with locomotor tests. Both series of experiments yielded similar results, despite notable differences in the PWTs measured, which can be explained by the differences of experimental conditions.

## Mechanical sensitivity

Mechanical sensitivity was assessed using Von Frey method. Mice were habituated to the testing environment before baseline testing. On the day of behavior testing, mice were placed in compartments on an elevated grid to allow access to the paw. Different filaments, ranging from 0.02 to 1.4 g, were applied perpendicularly to the plantar surface of the operated paw. 50% PWT was determined using an adaptation of the up and down method (*Chaplan et al., 1994*).

## Cold sensitivity

Cold sensitivity was assessed by immersing the operated paw in a 18°C water bath until withdrawal or shaking was observed. In order to avoid stressing mice that are manually tethered in a piece of tissue, they were habituated to the test for 3 days before the baseline test in room temperature water. The first two latencies measured with a difference of less than 2 s were averaged to obtain the pain withdrawal latency.

## Motor behaviors

Motor coordination assessments were performed using the rotarod test. Mice were placed on the rotarod device (Bioseb) during 3 min with an accelerating ramp of 4–6 rounds per min. The latency to fall was automatically measured. The test was performed three times for each day of measurement, with two baseline measurements and one measurement at the end of the neuropathic tests. Spontaneous locomotion was also estimated at the end of the neuropathic tests using the circular corridor test. Mice were placed in the cyclotron (IMetronic) in the dark. Four detectors located around the corridor allowed the measurement of the number of quarter turns performed by the animals during 2 hr.

In each mouse, immunolabeling of GFP was systematically performed at the end of the behavioral tests. Observation of mCherry and GFP expression allowed to control the localization/extend of the virus infection and the efficiency of Cav3.2 deletion in KO mice.

## Quantification and statistical analysis

No statistical methods were used to predetermine sample sizes, which are comparable to many studies using similar techniques and animal models.

## Imaging data analysis

Confocal images acquired for the evaluation of coexpression of Cav3.2-GFP, PV, and NeuN were processed using the Fiji/ImageJ software, with the Cell Counter plug-in for manual cell counts. Quantification of the co-labeling of GFP and NeuN were performed on three mice (two males, one female)

by analyzing 11–18 images in 5–8 brain slices per mouse. Quantification of the co-labeling of GFP and PV were performed on three mice (two males, one female) by analyzing 10–12 images in 5–8 brain slices per mouse. Data are expressed as mean ± standard deviation.

## Electrophysiological data analysis

For in vivo data analysis, potential single units were first identified using automated clustering software utilizing peak and trough feature sets (KlustaKwik). These clusters were then examined manually for waveform shape (SpikeSort3D, Neuralynx). Upon examination of the interspike intervals, multiunit clusters were discarded.

Bursts consisted of an initial spike that was followed by one or more spikes at an interval equal to or less than 5 ms. Since we were interested in potentially T-type channel-mediated bursts, another exclusion criterion was added: bursts had to be preceded by a pause in the spiking activity of at least 50 ms. Bursts recorded in response to the 470 nm blue-light pulse used for PINP were not included in the calculation of the spiking activity.

For in vivo and in vitro experiments, each neuron was considered as an independent observation since the number of recorded units/cells varied considerably from one mouse to another (2–6 units per mouse in the case of in vivo experiments and 1–3 neurons in the case of in vitro experiments). The number of mice and the average number of recorded units/neurons per mouse are indicated in the 'source data' files for each experimental condition.

Quantification and statistical analysis of in vivo and in vitro data were performed with the Igor Pro v6 and Matlab 2019b softwares, respectively. Between conditions comparison was based on Wilcoxon rank-sum test (Mann–Whitney $U$-test) and Wilcoxon signed rank test for unpaired and paired datasets, respectively. Differences were considered significant if the p value was lower than 0.05. All data were presented as the means ± standard deviation.

## Behavioral data analysis

Behavioral data analysis and statistics were performed in R (version 4.1.0), using the RStudio software. Data normality was checked using the Shapiro–Wilks test and no normal distribution was found throughout the datasets.

For mechanical and cold sensitivity, the presence of statistical differences in treatment across the multiple time points was assessed within each group (male KO, female KO, male KI, female KI) using the nonparametric Friedman test. The existence of significant differences between the two baseline measurements was checked using a paired Wilcoxon signed rank test. As no significant difference was found between baseline measurements within each group, these values were averaged as a mean baseline value. Each measurement performed after SNI surgery was then compared to the mean baseline value with the paired Wilcoxon signed rank test. Global intergroup comparisons (KI vs. KO) were performed using the Kruskal–Wallis test. At each time point, significant differences between groups were further identified using the Wilcoxon rank-sum test.

For the Rotarod test, the existence of significant differences within each group in the latency to fall measured in baseline conditions and 21 days after SNI surgery was assessed using the Wilcoxon signed rank test. Intergroup comparison was performed using the Wilcoxon rank-sum test.

For the circular corridor test, intergroup comparison was performed using the Wilcoxon rank-sum test.

Differences were considered significant if the p value was lower than 0.05. All values are expressed as mean ± standard error to the mean.

## Acknowledgements

This work was supported by operating grants from the Agence Nationale de la Recherche (ANR Pain-T and 'Investissements d'Avenir' program I-Site CAP 20-25).

# Additional information

## Funding

| Funder | Grant reference number | Author |
|---|---|---|
| Agence Nationale de la Recherche | ANR-15-CE16-0012-03 | Emmanuel Bourinet |

The funders had no role in study design, data collection, and interpretation, or the decision to submit the work for publication.

## Author contributions

Sophie L Fayad, Conceptualization, Formal analysis, Validation, Investigation, Visualization, Methodology, Writing – original draft; Guillaume Ourties, Formal analysis, Validation, Investigation, Visualization, Methodology; Benjamin Le Gac, Conceptualization, Formal analysis, Validation, Investigation, Visualization, Methodology; Baptiste Jouffre, Sylvain Lamoine, Antoine Fruquière, Sophie Laffray, Investigation; Laila Gasmi, Investigation, Methodology; Bruno Cauli, Emmanuel Bourinet, Conceptualization, Supervision, Validation, Methodology; Christophe Mallet, Supervision, Validation, Methodology; Thomas Bessaih, Conceptualization, Data curation, Formal analysis, Validation, Investigation, Visualization, Methodology, Writing – original draft; Régis C Lambert, Conceptualization, Data curation, Software, Formal analysis, Supervision, Funding acquisition, Validation, Investigation, Methodology, Writing – original draft, Project administration, Writing – review and editing; Nathalie Leresche, Conceptualization, Data curation, Formal analysis, Supervision, Funding acquisition, Validation, Investigation, Methodology, Writing – original draft, Project administration, Writing – review and editing

## Author ORCIDs

Sophie L Fayad ![ORCID] http://orcid.org/0000-0002-4834-6524
Bruno Cauli ![ORCID] http://orcid.org/0000-0003-1471-4621
Christophe Mallet ![ORCID] http://orcid.org/0000-0002-0873-6763
Thomas Bessaih ![ORCID] http://orcid.org/0000-0003-0764-7731
Régis C Lambert ![ORCID] http://orcid.org/0000-0001-8972-1151
Nathalie Leresche ![ORCID] http://orcid.org/0000-0001-6705-9769

## Ethics

All procedures complied with the ethical guidelines of the Federation for Laboratory Animal Science Associations (FELASA) and with the approval of the French National Consultative Ethics Committee for health and life sciences (authorization number: 17958).

## Decision letter and Author response

Decision letter https://doi.org/10.7554/eLife.79018.sa1
Author response https://doi.org/10.7554/eLife.79018.sa2

---

# Additional files

## Supplementary files

• MDAR checklist

## Data availability

All data generated or analyzed during this study are included in the manuscript and supporting files.

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

# Appendix 1

## Appendix 1—key resources table

| Reagent type (species) or resource | Designation | Source or reference | Identifiers | Additional information |
|---|---|---|---|---|
| Strain, strain background (*Mus musculus*, male and female) | Cav3.2-eGFPflox | PMID:25600872 (*François et al., 2015*) | Cacna1h<sup>tm1.1(epH)Ebo</sup>/J | KI mice with two *loxP* site-flanked ecliptic-GFP tag into exon 6 of the *Cacna1h* gene |
| Strain, strain background (*Mus musculus*, female) | PV-Cre | Jackson lab | B6.129P2-Pvalb-<sup>tm1(cre)Arbr</sup>/J | Cre recombinase expression in *Pvalb*-expressing cells |
| Strain, strain background (*Mus musculus*, male) | Ai14 | Jackson lab | B6.Cg-Gt(ROSA) 26Sor-<sup>tm14(CAG-tdTomato)Hze</sup>/J | Cre-dependent expression of the red fluorescent protein variant (tdTomato) |
| Strain, strain background (*Mus musculus*, male) | Ai32 | Jackson lab | B6.Cg-Gt(ROSA) 26Sor<sup>tm32(CAG-COP4*H134R/EYFP)Hze</sup>/J | Cre-dependent expression of the ChR2(H134R)-EYFP |
| Sequence-based reagent | Mouse *Pvalb* external sense | PMID:21795545 (*Tricoire et al., 2011*) | PCR primers | GCCTGAAGAAAAAGAACCCG |
| Sequence-based reagent | Mouse *Pvalb* external antisense | PMID:21795545 (*Tricoire et al., 2011*) | PCR primers | AATCTTGCCGTCCCCATCCT |
| Sequence-based reagent | Mouse *Pvalb* internal sense | PMID:21795545 (*Tricoire et al., 2011*) | PCR primers | CGGATGAGGTGAAGAAGGTGT |
| Sequence-based reagent | Mouse *Pvalb* internal antisense | PMID:21795545 (*Tricoire et al., 2011*) | PCR primers | TCCCCATCCTTGTCTCCAGC |
| Sequence-based reagent | Mouse *Gad2* external sense | PMID:34766906 (*Karagiannis et al., 2021*) | PCR primers | CCAAAAGTTCACGGGCGG |
| Sequence-based reagent | Mouse *Gad2* external antisense | PMID:34766906 (*Karagiannis et al., 2021*) | PCR primers | GTGAGCAGTATCGCAGCCCC |
| Sequence-based reagent | Mouse *Gad2* internal sense | PMID:34766906 (*Karagiannis et al., 2021*) | PCR primers | CACCTGCGACCAAAAACCCT |
| Sequence-based reagent | Mouse *Gad2* internal antisense | PMID:12196560 (*Férézou et al., 2002*) | PCR primers | GATTTTGCGGTTGGTCTGCC |
| Sequence-based reagent | Mouse *Gad1* external sense | PMID:23565079 (*Cabezas et al., 2013*) | PCR primers | TACGGGGTTCGCA CAGGTC CGGGCGG |
| Sequence-based reagent | Mouse *Gad1* external antisense | PMID:23565079 (*Cabezas et al., 2013*) | PCR primers | CCCAGGCAGCATCCACAT |
| Sequence-based reagent | Mouse *Gad1* internal sense | PMID:23565079 (*Cabezas et al., 2013*) | PCR primers | CCCAGAAGTGAAGACAAAAGGC |
| Sequence-based reagent | Mouse *Gad1* internal antisense | PMID:23565079 (*Cabezas et al., 2013*) | PCR primers | AATGCTCCGTAAACAGTCGTGC |
| Sequence-based reagent | Mouse *Slc17a6* external sense | This paper | PCR primers | TGGAGAAGAAGCAGGACAACC |
| Sequence-based reagent | Mouse *Slc17a6* external antisense | This paper | PCR primers | GTGAGCAGTATCGCAGCCCC |
| Sequence-based reagent | Mouse *Slc17a6* internal sense | This paper | PCR primers | TGACAGAGGACGGTAAGCCCC |
| Sequence-based reagent | Mouse *Slc17a6* internal antisense | This paper | PCR primers | TCATCCCCACGGTCTCGG |
| Sequence-based reagent | Mouse *Cacna1g* external sense | This paper | PCR primers | CACCGATGTCACTGCCCAAG |
| Sequence-based reagent | Mouse *Cacna1g* external antisense | This paper | PCR primers | GGCTCTCCTGACCCTCTCCA |
| Sequence-based reagent | Mouse *Cacna1g* internal sense | This paper | PCR primers | GCTCTCGCCGCACCAGTA |
| Sequence-based reagent | Mouse *Cacna1g* internal antisense | This paper | PCR primers | CTTGGGCTCCTACGCTTCAG |

*Appendix 1 Continued on next page*

*Appendix 1 Continued*

| Reagent type (species) or resource | Designation | Source or reference | Identifiers | Additional information |
|---|---|---|---|---|
| Sequence-based reagent | Mouse *Cacna1h* external sense | This paper | PCR primers | TACCAGACAGAGGAGGGCGA |
| Sequence-based reagent | Mouse *Cacna1h* external antisense | This paper | PCR primers | CTATCACCACCAGGCACAGG |
| Sequence-based reagent | Mouse *Cacna1h* internal sense | This paper | PCR primers | CATTCATCTGCTCCTCACGC |
| Sequence-based reagent | Mouse *Cacna1h* internal antisense | This paper | PCR primers | GCCCACAATGATGAGGAGGA |
| Sequence-based reagent | Mouse *Cacna1i* external sense | This paper | PCR primers | GTCCCCCTCCATCCCCTC |
| Sequence-based reagent | Mouse *Cacna1i* external antisense | This paper | PCR primers | CAATGAAGAAGTCCAAGCGGTT |
| Sequence-based reagent | Mouse *Cacna1i* internal sense | This paper | PCR primers | GTTGCCTTCTTCTGCCTGCG |
| Sequence-based reagent | Mouse *Cacna1i* internal antisense | This paper | PCR primers | TCCCCGAGGTAGCACTTCTT |
| Strain, strain background (*AAV*) | AAV8-hSyn-mCherry-Cre | UNC Vector Core | | |
| Strain, strain background (*AAV*) | AAV8-hSyn-mCherry | UNC Vector Core | | |
| Antibody | Anti-GFP (chicken polyclonal) | Life Technologies | A10262 | 1:500 |
| Antibody | Anti-GFP (rabit polyclonal) | Chromotek | PABG1 | 1:500 |
| Antibody | Anti-Parvalbumin (mouse monoclonal) | Sigma-Aldrich | P3088 | 1:1000 |
| Antibody | Anti-NeuN (rabbit polyclonal) | Merck Millipore | ABN78 | 1:500 |
| Antibody | Anti-chicken-Alexa Fluor 488 (goat polyclonal) | Life Technologies | A11039 | 1:500 |
| Antibody | Anti-chicken-Alexa Fluor 488 (donkey polyclonal) | Sigma-Aldrich | SAB4600031 | 1:500 |
| Antibody | Anti-rabit-Alexa Fluor 647 (goat polyclonal) | Life Technologies | A21244 | 1:500 |
| Antibody | Anti-mouse-Alexa Fluor 488 (goat polyclonal) | Life Technologies | A11001 | 1:500 |
| Antibody | Anti-rabit-Cyanine Cy3 (donkey polyclonal) | Jackson ImmunoResearch Europe | 711-165-1525 | 1:2000 |
| Chemical compound, drug | Isoflurane | Iso-Vet | Cat# 3248850; GTIN: 18904026625157 | |
| Chemical compound, drug | Buprenorphine (Buprecare) | Axience | GTIN: 03760087151893 | |
| Chemical compound, drug | Pentobarbital (Euthasol Vet) | Dechra | GTIN: 08718469445110 | |
| Chemical compound, drug | Ketamine | Imalgene 1000 | GTIN: 03661103003199 | |
| Chemical compound, drug | Xylazine (Rompun) | Bayer | GTIN: 04007221032311 | |
| Chemical compound, drug | Bupivacaine | Henry Schein | Cat# 054879 | |
| Chemical compound, drug | Twelve TVM Eye Support Drops | TVM UK Animal Health | GTIN: 03700454507502 | |
| Chemical compound, drug | Vetedine | Vetoquinol | GTIN: 03605870001385 | |

*Appendix 1 Continued on next page*

*Appendix 1 Continued*

| Reagent type (species) or resource | Designation | Source or reference | Identifiers | Additional information |
|---|---|---|---|---|
| Chemical compound, drug | NaCl | Sigma-Aldrich | Cat# S6191; CAS: 7647-14-5 | |
| Chemical compound, drug | KCl | Sigma-Aldrich | Cat# 60128; CAS: 7447-40-7 | |
| Chemical compound, drug | $CaCl_2$ | Sigma-Aldrich | Cat# C3881; CAS: 10035-04-8 | |
| Chemical compound, drug | $MgCl_2$ | Sigma-Aldrich | Cat# M2670; CAS: 7791-18-6 | |
| Chemical compound, drug | $NaH_2PO_4$ | Sigma-Aldrich | Cat# S5011; CAS: 7558-80-7 | |
| Chemical compound, drug | $NaHCO_3$ | Sigma-Aldrich | Cat# 31437; CAS: 144-55-8 | |
| Chemical compound, drug | Glucose | Sigma-Aldrich | Cat# 49159; CAS: 14431-43-7 | |
| Chemical compound, drug | Potassium gluconate | Sigma-Aldrich | Cat# G4500; CAS: 299-27-4 | |
| Chemical compound, drug | EGTA | Sigma-Aldrich | Cat# E3889; CAS: 67-42-5 | |
| Chemical compound, drug | HEPES | Sigma-Aldrich | Cat# H3375; CAS: 7365-45-9 | |
| Chemical compound, drug | Disodium ATP | Sigma-Aldrich | Cat# A7699; CAS: 34369-07-8 | |
| Chemical compound, drug | Biocytin | Sigma-Aldrich | Cat# B4261; CAS: 576-19-2 | |
| Chemical compound, drug | D-Mannitol | Sigma-Aldrich | Cat# M9647; CAS: 69-65-8 | |
| Chemical compound, drug | Sodium Pyruvate | Sigma-Aldrich | Cat# P2256; CAS: 113-24-6 | |
| Chemical compound, drug | Kynurenic acid | Hello Bio | Cat# HB0363; CAS: 2439-02-3 | |
| Chemical compound, drug | Paraformaldehyde in 0.1 M phosphate buffer saline | Electron Microscopy Sciences | Cat# 15710; CAS: 30525-89-4 | |
| Chemical compound, drug | Trizma | Sigma-Aldrich | Cat# T1503; CAS: 77-86-1 | |
| Chemical compound, drug | Sodium azide | Sigma-Aldrich | Cat# S2002; CAS: 26628-22-8 | |
| Chemical compound, drug | Phosphate buffer saline | Sigma-Aldrich | Cat# D1408 | |
| Chemical compound, drug | Triton X-100 | Sigma-Aldrich | Cat# T8787; CAS: 9002-93-1 | |
| Chemical compound, drug | Fluoromount Aquaous Mounting Medium | Sigma-Aldrich | Cat# F4680 | |
| Chemical compound, drug | Donkey serum | Sigma-Aldrich | Cat# D9663 | |
| Chemical compound, drug | Streptavidin-Rhodamine-RedX | Jackson ImmunoResearch Europe | Cat# 016-290-084 | |
| Chemical compound, drug | DiI, 10% in ethanol | Invitrogen | Cat# V22885 | |
| Chemical compound, drug | CNQX | Hello bio | Cat# HB02057; CAS: 479347-85-8 | |
| Chemical compound, drug | SR95531 hydrobromide | Hello bio | Cat# HB0901; CAS: 104104-50-9 | |
| Chemical compound, drug | TTX | Latoxan | Cat# L8503; CAS: 4368-28-9 | |

*Appendix 1 Continued on next page*

*Appendix 1 Continued*

| Reagent type (species) or resource | Designation | Source or reference | Identifiers | Additional information |
|---|---|---|---|---|
| Chemical compound, drug | Nickel chloride hydrate | Sigma-Aldrich | Cat# 364304; CAS: 69098-15-3 | |
| Chemical compound, drug | TTA-P2 | Alomone labs | Cat# T-155; CAS: 918430-49-6 | |
| Chemical compound, drug | Taq DNA Polymerase | Qiagen | 201205 | |
| Chemical compound, drug | RNasin Ribonuclease Inhibitors | Promega | N2511 | |
| Chemical compound, drug | SuperScript II Reverse Transcriptase | Invitrogen | Cat# 18064014 | |
| Software, algorithm | Pclamp V 10.2 | Molecular Devices | https://www.molecular devices.com/ | |
| Software, algorithm | Matlab 2019b | MathWorks | https://fr.mathworks.com/ | |
| Software, algorithm | Igor Pro V6 | Wavemetrics | https://www.wavemetrics.com | |
| Software, algorithm | Cheetah V 5 | Neuralynx | https://neuralynx.com/software/cheetah | |
| Software, algorithm | KlustaKwik | Neuralynx | https://neuralynx.com/software/cheetah | |
| Software, algorithm | SpikeSort3D V 2 | Neuralynx | https://neuralynx.com/software/cheetah | |
| Software, algorithm | Fiji/ImageJ | NIH | https://imagej.nih.gov/ij/download.html | |
| Software, algorithm | Rstudio | | https://www.rstudio.com/ | |
| Software, algorithm | R version 4.1.0 (2021-05-18) | Camp Pontanezen – The R Foundation for Statistical Computing | https://www.r-project.org/ | |
| Other | Quartz-insulated platinum/tungsten (90%/10%) tetrodes | Thomas Recording GmbH | Cat# AN000259 | See 'In vivo electrophysiological recordings' in the Method Details |
| Other | Optical fibers | Thomas Recording GmbH | Cat# AN000514 | Optical fiber used to deliver 470-nm blue-light pulse for Photo-assisted Identification of Neuronal Population |
| Other | Borosilicate glass capillaries | Hilgenberg | Cat# 1409250 | See 'In vitro whole-cell patch-clamp recording' in the Method Details |
| Other | 4-0 Vicryl | Ethicon | ref JV397 | See 'Virus stereotaxic injections' in the Method Details |

