## [Editor Report]

The manuscript shows an important role for Cav2.3 channels in SNI-mediated allodynia and the firing properties of PV-expressing APT neurons. Mechanisms that underlie adaptations in chronic pain models are extremely important for the development of novel therapeutics for chronic pain and this could be a significant contribution in that regard.

---

## [Decision Letter]

**Decision letter after peer review:**

Thank you for submitting your article "Centrally expressed Cav3.2 T-type calcium channel is critical for the initiation and maintenance of neuropathic pain" for consideration by *eLife*. Your article has been reviewed by 3 peer reviewers, and the evaluation has been overseen by a Reviewing Editor and Gary Westbrook as the Senior Editor. The following individuals involved in the review of your submission have agreed to reveal their identity: Susan Ingram (Reviewer #1); Slobodan Todorovic (Reviewer #3).

Essential revisions:

1. Although there is considerable support for the findings, there is consensus among the reviewers that it is essential to provide direct evidence that the SNI increase in firing involved Cav3.2 channels that you characterized. Specifically, it is important to show that Ni-sensitive currents and rebound bursts in APT neurons are altered by SNI and that they are diminished/attenuated by APT KO. Addressing this question would examine the behavior in the SNI model in the KO, or in vitro recordings from the KO should be added.

2. The above is the major concern required in a revision. But, in addition, appropriate statistics that use a nested design for multiple cells in animals should be used for Figure 2.

3. Please also attend to other issues raised in the original comments of the reviewers below.

We think these studies can be performed within two months (the usual period allowed for *eLife* revisions) and look forward to seeing a suitably revised version. Please note, however, that if you are not able to demonstrate that Cav3.2 channels indeed contribute to SNI-induced changes in burst firing, it will not be publishable in *eLife*.

*Reviewer #1 (Recommendations for the authors):*

The recommendation is to provide details of the statistics used and rationale for analysis of multiple observations in single animals. It is not clear if they are treated as independent observations or corrected for in the analyses.

*Reviewer #2 (Recommendations for the authors):*

1. Figure 1, does SNI alter the expression of Cav3.2 and the overlap between Cav3.2 and PV in APT?

2. Figure 2, please clarify whether light-evoked bursts are included when calculating the mean firing rate? If yes, what's the mean firing rate in the absence of evoked bursts, and is it different between naïve and SNI mice? Does SNI alter mean firing and/or evoked bursts in the ipsilateral APT? Sham mice should be used as the control of SNI.

3. Figure 3, are the amplitude of Ni-sensitive current, the number of spikes in rebound burst and the spike-current relationship altered by SNI? Are they attenuated in APT-KO mice after sham/SNI?

4. Figure 4B-C, in SNI mice that receive AAV-Cre, does blocking peripheral/spinal Cav3.2 further attenuate/normalize mechanical allodynia?

---

## [Author Response]

Reviewer #1 (Recommendations for the authors):The recommendation is to provide details of the statistics used and rationale for analysis of multiple observations in single animals. It is not clear if they are treated as independent observations or corrected for in the analyses.

We considered each unit as an independent observation. We did so since the number of recorded PV+-units per animal (identified with the PINP method) was small and varied greatly between animals, from 2 to 6 units. We are not aware of statistical methods using a nested design for multiple cells in animals that could be used in such condition.

Since the measured variables did not follow normal distributions, we performed unpaired comparison with the Wilcoxon sum rank test. This is now stated in the section ‘quantification and statistical analysis’ (page 22, line 680). P-values are now included in the figures and the result section when appropriate.

Reviewer #2 (Recommendations for the authors):1. Figure 1, does SNI alter the expression of Cav3.2 and the overlap between Cav3.2 and PV in APT?

We did not perform a systematic comparison of the Cav3.2 and PV colocalization between control and SNI mice but we did not observe a change in the number of GFP-expressing neurons per slice after SNI.

2. Figure 2, please clarify whether light-evoked bursts are included when calculating the mean firing rate? If yes, what's the mean firing rate in the absence of evoked bursts, and is it different between naïve and SNI mice? Does SNI alter mean firing and/or evoked bursts in the ipsilateral APT? Sham mice should be used as the control of SNI.

Light-evoked bursts were not included when calculating the mean firing rate. We apologize for this omission in the description of the analysis protocol. This point is now specified in the Materials and methods section (page 22, line 677).

We did not record in the ipsilateral APT.

We fully agree that we should have used sham animals in first instance and we accordingly performed a new set of experiments. Now, all control animals underwent sham surgery. Statistically significant differences between sham operated and SNI animals were observed similarly to what was previously reported for the naïve/SNI comparison (see new Figure 2 and Results section page 3, lines 98-116).

3. Figure 3, are the amplitude of Ni-sensitive current, the number of spikes in rebound burst and the spike-current relationship altered by SNI? Are they attenuated in APT-KO mice after sham/SNI?

We could not directly assess the impact of SNI on Cav3.2 current amplitude for technical reasons. Indeed, at the beginning of our study, we tried to record low-threshold calcium currents in APT neurons. Although we choose to record these currents in very young mice (P12 -14) to minimize the space clamp problem, it was clearly impossible to clamp the membrane potential even at this age when the dendritic arborization is not fully mature. Many experimental and theoretical studies have shown that in non-spherical cells, voltage-dependent currents, such as Cav3, are severely distorted due to the lack of space clamp (see for example Bar-Yehuda and Korngreen J. neurophysiol 2008). Recording Ni^2+^ sensitive currents in sham versus SNI mice, approximately 100 days old, cannot be performed with the minimal accuracy required for rigorous comparison.

Therefore, we performed a new set of experiments by recording rebound burst responses in PV+ APT neurons in sham and SNI mice. By comparing the rebound bursts, we observed that the distribution of the maximum number of spikes in the rebound bursts is shifted to larger values in SNI compared to sham mice. To evaluate whether Cav3.2 channels are indeed responsible for this increase in bursts, we applied 100*µM* Ni^2+^ and observed that this difference in distribution disappeared. These results, which clearly show a Cav3.2 related enhance rebound burst in APT neurons after SNI are now presented page 4, line 150-158, in the new figure 4 (panel D) and outlined as one of the main result of the present study at the end of the introduction (page 2, line 71).

We could not perform similar experiments in APT-KO mice. Given that Cav3.2-expressing neurons represent 20% of the APT neuron population, comparing bursting activity in KI and APT-KO mice after sham/SNI would require identification of this specific subpopulation in living tissue, which is clearly not possible in APT-KO mice. As far as KI mice are concerned, although a GFP tag is attached to the Cav3.2 channels in the Cav3.2^eGFP-flox^ knock-in, its fluorescence is not detectable without antibody preventing direct identification in living tissue (François et al., 2015).

4. Figure 4B-C, in SNI mice that receive AAV-Cre, does blocking peripheral/spinal Cav3.2 further attenuate/normalize mechanical allodynia?

At the peripheral level, deletion of Cav3.2 in a subset of sensory neurons altered light-touch perception and noxious mechanical, cold and chemical sensation (François et al., Cell Reports, 2015). In the SNI model, such deletion also induced a marked reduction in the mechanical allodynia similar to that reported here. To date, no data are available regarding the effect of a specific Cav3.2 KO at the spinal level. Indeed, the multiple sites where Cav3.2 seems to play a crucial role in chronic pain open very interesting question about the respective roles of centrally versus peripherally expressed Cav3.2 channels in allodynia. However, such studies would require a large number of experimental investigations that are well beyond the scope of the present work.